# Laboratory Investigation of Hybrid IgG4 k/λ in MuSK Positive Myasthenia Gravis

**DOI:** 10.3390/ijms22179142

**Published:** 2021-08-24

**Authors:** Umberto Basile, Cecilia Napodano, Francesca Gulli, Krizia Pocino, Riccardo Di Santo, Laura Todi, Valerio Basile, Carlo Provenzano, Gabriele Ciasca, Mariapaola Marino

**Affiliations:** 1Dipartimento di Scienze di Laboratorio e Infettivologiche, Fondazione Policlinico Universitario “A. Gemelli” IRCCS, Università Cattolica del Sacro Cuore, 00168 Rome, Italy; umberto.basile@policlinicogemelli.it; 2Synlab Data Medica, 35133 Padova, Italy; cecilia.napodano@gmail.com; 3Laboratorio di Patologia Clinica, Ospedale Madre Giuseppina Vannini, 00177 Rome, Italy; dottfgulli@gmail.com; 4Unità Operativa Complessa Patologia Clinica, Ospedale Generale di Zona, San Pietro Fatebenefratelli, 00189 Rome, Italy; krizia.pocino@gmail.com; 5Dipartimento di Neuroscienze, Sezione di Fisica, Fondazione Policlinico “A. Gemelli” IRCCS, Università Cattolica del Sacro Cuore, 00168 Rome, Italy; riccardo.disanto92@gmail.com (R.D.S.); gabriele.ciasca@unicatt.it (G.C.); 6Dipartimento di Medicina e Chirurgia Traslazionale, Sezione di Patologia Generale, Fondazione Policlinico Universitario “A. Gemelli” IRCCS, Università Cattolica del Sacro Cuore, 00168 Rome, Italy; laura.todi@unicatt.it (L.T.); carlo.provenzano@unicatt.it (C.P.); 7Clinical Pathology Unit and Cancer Biobank, Department of Research, Advanced Diagnostics and Technological Innovation, IFO-Regina Elena National Cancer Institute, 00128 Rome, Italy; valeriobasile90@gmail.com

**Keywords:** biomarker, antibodies, personalized medicine, IgG4, Fab-arm exchange, hybrid k/λ, MuSK-MG

## Abstract

Myasthenia gravis with antibodies (Abs) against the muscle-specific tyrosine kinase (MuSK) is a rare autoimmune disorder (AD) of the neuromuscular junction (NMJ) and represents a prototype of AD with proven IgG4-mediated pathogenicity. Thanks to the mechanism of Fab-arm exchange (FAE) occurring in vivo, resulting MuSK IgG4 k/λ Abs increase their interference on NMJ and pathogenicity. The characterization of hybrid MuSK IgG4 as a biomarker for MG management is poorly investigated. Here, we evaluated total IgG4, hybrid IgG4 k/λ, and the hybrid/total ratio in 14 MuSK-MG sera in comparison with 24 from MG with Abs against acetylcholine receptor (AChR) that represents the not IgG4-mediated MG form. In both subtypes of MG, we found that the hybrid/total ratio reflects distribution reported in normal individuals; instead, when we correlated the hybrid/total ratio with specific immune-reactivity we found a positive correlation only with anti-MuSK titer, with a progressive increase of hybrid/total mean values with increasing disease severity, indirectly confirming that most part of hybrid IgG4 molecules are engaged in the anti-MuSK pathogenetic immune-reactivity. Further analysis is necessary to strengthen the significance of this less unknown biomarker, but we retain it is full of a diagnostic-prognostic powerful potential for the management of MuSK-MG.

## 1. Introduction

A different profile of serological IgG subclasses has been widely described in patients (pts) with autoimmune disorders (AD), increasing the interest to properly understand their contribution in pathogenesis and clinical significance for a different personalized management of disease [1]. IgG4 subclass represents the least frequent among IgG subclasses (about 5% of total IgG) and develops after prolonged activation of the immune system with or without antigen [2]. Differently from IgG1 and IgG3 subclasses that bind two identical antigen epitopes cross-linked by the two identical Fabs, after secretion into the plasma, IgG4 can swap half-molecules (one heavy-chain with one attached light chain), with other IgG4 half-molecules (Figure 1, panels A and B).

This process has been named “Fab-arm exchange” (FAE) and results in hybrid k/λ antibodies (Abs), bispecific (because composed of two different binding sites) and functionally monovalent for each antigen recognized [3]. Due to this mechanism, IgG4 molecules fail to cross-link membrane-bound identical antigens (without endocytosis) and to form immune complexes on antigen expressing cells (without complement activation); IgG4 Abs, therefore, are generally considered non-pathogenic with a more protective role in inflammatory reactions and allergy [2,3]. Spontaneous FAE in vivo takes account of a large proportion of bispecific IgG4s in normal human serum. This process can be induced in vitro under reducing conditions as well [4].

The capacity of hybrid IgG4 molecules to interfere with the antigen function has been investigated in well-recognized IgG4-mediated models of diseases. A prototype of this nosographic entity is the subtype of Myasthenia gravis (MG) with Abs against the muscle-specific tyrosine kinase receptor (MuSK), a transmembrane protein of the agrin receptor complex involved in acetylcholine receptor (AChR) clustering as well as in organizing and maintaining the neuromuscular junction (NMJ) [5]. MG is a rare AD of the neuromuscular transmission, with an annual incidence rate of 5.3 per million person-years and a prevalence of 77.7 cases per million that is increased thanks to the improvement on diagnostics and clinical care therapy [6]. While more than 80% of patients with generalized disease display Abs against the AChR at the NMJ, up to 7% of them display anti-MuSK Abs that, differently from the AChR-subtype of MG, belong to the IgG4 subclass for the most part [7,8,9]. Anti-MuSK IgG4 can inhibit agrin-induced MuSK activation binding to the extracellular Ig1-like domain, which is crucial for the signal transduction pathway required for the clustering of AChR molecules [10]. The resulting impairment of neuromuscular transmission underlies the clinically severe and fatigable weakness of skeletal muscles. It has been demonstrated that MuSK IgG4s undergo FAE in vitro and in vivo, but without affecting their pathogenicity: while monospecific bivalent anti-MuSK Abs would have a protective effect by enhancing MuSK phosphorylation, functional monovalency following FAE increases their interfering effect on AChR clustering and therefore their pathogenicity [11,12,13]. Starting from the proven pathogenicity of hybrid MuSK IgG4s, the open question is to define if their characterization in diagnostics may add a plus value in the management of disease. In this paper, we reported our results of a pilot serological analysis aimed to explore the levels of hybrid IgG4 molecules in MuSK-MG, in comparison with AChR-MG used as a not IgG4-mediated MG control.

## 2. Results

### 2.1. Total and Hybrid IgG4 k/λ Levels in MG Patients

A box plot analysis of total IgG4, hybrid IgG4 k/λ, and the hybrid/total ratio is visualized in Figure 2, together with the results of a Wilcoxon Unpaired two-sample test; data are reported as median [IQR] in Table 1. For this evaluation, we analyzed serum samples from 14 MuSK-MG and 24 AChR-MG pts.

Although there is an increase in median values, the statistical analysis revealed no significant differences in total IgG4 levels between MuSK-MG and AChR-MG patients (547.92 [646.78] mg/L vs. 267.73 [366.81] mg/L, respectively), and hybrid IgG4 levels (195.39 [174.39] mg/L vs. 87.16 [82.81] mg/L, for MuSK and AChR respectively). The hybrid/total IgG4 ratio that we found (0.37 [0.13] and 0.38 [0.11] for MuSK- and AChR-MG respectively) reflects the normal distribution of ~30% described in normal human sera (NHS) [4]. 

### 2.2. Correlation between the Hybrid/Total IgG4 Ratio and Specific MG Immune-Reactivity

Laboratory determinations, along with the MG Foundation of America (MGFA) clinical classification of disease recorded at the time of collection, are reported in Appendix A for MuSK-MG pts, and in Appendix A for AChR-MG. We evaluated the relationship between hybrid/total IgG4 ratio with the specific Abs titer in MuSK- and AChR-MG, visualized in Figure 3A,B respectively. For this purpose, a linear model was fitted to data and the corresponding regression line is shown together with confidence and prediction bands. Fitting residues were also visualized, and normality assessed with a Shapiro–Wilk test, showing no deviation from normality in Figure 3A (*p* = 0.32) and Figure 3B (*p* = 0.18). The following intercepts (q) and slopes (m) were measured, namely q = (0.22 ± 0.17) ^ns^ nmol/L and m = (1.91 ± 0.48) ** nmol/L for MuSK-MG, and q = (15.3 ± 3.2) *** nmol/L and m = (−12.8 ± 7.4) ^ns^ nmol/L for AChR-MG. 

We noticed that a positive and significant slope is measured only for MuSK-MG pts (Figure 3A), while m is consistent with 0 within two standard deviations for AChR-MG pts (Figure 3B). Consistently, in Figure 3C we report the Pearson correlation coefficient for both datasets, obtaining a strong positive and significant correlation only for MuSK-MG (ρ = 0.77, *p* = 0.002) while no significant negative correlation for AChR-MG pts (ρ = −0.35, *p* = 0.098). 

### 2.3. Correlation between Hybrid/IgG4 and MGFA Score in MuSK-MG

To verify if hybrid IgG4 plays a pathogenic role as already reported [11,12,13], we analyzed the correlation between hybrid/total IgG4 ratio and disease severity. In Figure 4, we visualized the distribution of Hybrid/Total IgG4 ratios at different MGFA scores for 13 out of 14 MuSK-MG pts who were evaluated for the MGFA at the time of withdrawal. As shown, despite the very limited number of samples prevents us from any statistical consideration, we noticed a progressive increase of the titer with the increasing of MGFA score. Specifically, IIb MuSK-MG pts had the lowest mean ratio of 0.20, IIIb pts an intermediate mean value of 0.35, and IVb pts showed the highest mean value of 0.43. These results are suggestive that the hybrid/IgG4 ratio correlates with disease severity, but this finding deserves a better in-depth study in a larger sample size to be confirmed. 

## 3. Discussion

MG represents the prototype of a rare and relatively unknown organ-specific antibody-mediated AD but also of a long-term chronic neuromuscular condition, characterized by skeletal muscle weakness and fatigue, as well as respiratory impairment in severe forms. With an annual estimated prevalence of 1.9–2.9 per million, MuSK-MG probably represents the prototype AD for whom the IgG4-mediated pathogenicity was well proven [14]. IgG4 subclass involvement has been investigated in a wide range of AD since the frequent increase of serum IgG4 levels associated to tissue infiltration with IgG4+ plasma cells and storiform fibrosis depicted a new identified group of IgG4-related disease (IgG4-RD) [15]. However, MuSK-MG does not fulfill IgG4-RD criteria, even if a recent case report opens a new scenario on a putative link between the two diseases [16]. In recent years, it has been well demonstrated that a considerable proportion of MuSK IgG4 is Fab-arm exchanged without affecting pathogenetic mechanisms of disease [11,12,13]. 

The FAE phenomenon naturally occurs in vivo taking account of bispecific IgG4s also in normal human serum. Data from the literature showed that hybrid/total IgG4 ratio in NHS is assessed at approximately 30% [4,17]. Analyses on serum samples from a range of IgG4-related AD were also conducted giving different results: a lower hybrid/total IgG4 ratio was described in samples from Pemphigus/Pemphigoid and Autoimmune Pancreatitis and a ratio similar to NHS in Primary Sclerosing Cholangitis [17].

Here, we show our findings of a pilot laboratory investigation of hybrid IgG4 k/λ assessment in MuSK-MG sera. We observed that, although an increase (not significant) in IgG4 total levels, the hybrid/total IgG4 ratio of 0.37 [0.13] is comparable to AChR-MG, the not IgG4-mediated form (0.38 [0.11]), and reflects the occurrence reported in the literature for NHS. Since only MuSK-MG is the IgG4-mediated MG form, we suggest that is not the total amount of hybrids per se the matter of the fact (and if it is accordant or discordant from NHS levels) but if hybrids (that are exclusively IgG4 molecules) are engaged in pathogenetic immune-reactivity or not. For this evaluation we retain more useful the comparison between the two MG subtypes, the hybrid IgG4 mediated (MuSK-MG) and the not hybrid IgG4-mediated (AChR-MG). When we compared the hybrid/total IgG4 ratios with anti-MuSK Abs titers we found a strong significant correlation (ρ = 0.77, *p* = 0.002) indirectly confirming that the most part of hybrid IgG4 molecules are engaged in the anti-MuSK immune reactivity, mediating pathogenicity (Figure 3A). Instead, when we analyzed the hybrid/total IgG4 ratios in relation to anti-AChR titers we found an inverse negative correlation, that is coherent with the assumption that AChR-MG is not an IgG4 mediated subtype of MG (Figure 3B). The observation that the mean value of hybrid/total IgG4 ratio in MuSK-MG increases along with disease severity according to the MGFA score further reinforces our previous result (Figure 4). Of course, a deeper evaluation on an enlarged group of pts together with complete information regarding all clinical correlates is necessary to validate this finding.

A substantial issue in the management of MG has always been represented by the occurrence of pts with myasthenic symptoms but that give negative results when Abs are studied. Since 2000, thanks to an extensive laboratory research, we gave a considerable contribution to clinicians for the identification of anti-MuSK Abs as identifiers of a new MG-subtype, different from AChR-MG, that, however, do not cover all the AChR negative pts [18]. We also concurred to describe other immune-reactivities reported in a minor proportion of MG patients, that can be co-existing with anti-AChR Abs and/or anti-MuSK Abs, targeting acetylcholinesterase [19] and the low-density lipoprotein receptor-related protein 4 LRP4 [20] with a contribution to improve the management of disease. However, MG diagnosis may remain puzzling in any cases. For this reasons, additional serological biomarkers are currently widely investigated. To find a measure of disease activity, many groups focused on the analysis of pro-inflammatory cytokines and molecules [21]. In this context, we recently reported the first evidence that MG pts display increased circulating level of free immunoglobulin light chains (FLC), describing different profiles in AChR- and MuSK-MG [22]. Considered as an emerging direct biomarker of B cell activity in AD, FLC has been confirmed as a challenging biomarker for MG in suspected pts who are double seronegative [23]. Since the demonstration that hybrid IgG4 mediates pathogenesis in MuSK-MG [11,12,13], our interest was focused on understanding the clinical relevance of such molecules.

A crucial point that we must consider regarding MG laboratory diagnostics is that the most validated and employed commercial radioimmunoassay (RIA) for the routine detection of specific auto-Abs is based on the immunoprecipitation of the radiolabeled relevant antigen (AChR or MuSK): the presence of monovalent Abs (how hybrids actually are) can partially impair this process. In contrast, in cell-based assays (CBA), which partially overcomes this problem, hybrid molecules do not affect test sensitivity. Therefore, the increase of hybrid IgG4 molecules in MuSK-MG pts can account for the relatively large number of RIA-negative CBA-positive results found in MuSK-MG sera [24].

Traditionally, AD have been treated with potent immunosuppressive drugs that, together with a reduction in mortality and morbidity, unfortunately cause a non-specific immune suppression with several potential adverse effects. In recent years, the advent of biological therapy widely modified treatment of a wide spectrum of AD [25]. Rituximab (RTX) has been increasingly emerged for the treatment of AD to deplete B lymphocytes and decrease autoantibody formation with beneficial results in B and T cell-mediated diseases [26]. RTX has proved particularly effective in the treatment of IgG4-related disease as well as in IgG4-mediated autoimmune conditions [27,28]. A more pronounced and long-standing response to RTX-induced B-cell depletion was observed in MuSK-MG, refractory to traditional therapies: it involves the circuit of specific anti-MuSK IgG4 as we previously demonstrated by the paralleled prolonged reduction of specific IgG4 MuSK antibodies [9]. Since most of pathogenetic MuSK IgG4 consists in hybrid molecules, our results could be very useful for a better selection of MuSK-MG pts to be treated with RTX [11,12,13].

We are conscious about the limits of our study. As it is a rare disease and due to the complex procedure employed currently not available as routine procedure, the laboratory analysis has been conducted on a small sample size of MuSK-MG sera from pts referred in past years to our Institution: unfortunately, many clinical data are not available or lost. However, our main goal is the effort to translate into the clinical practice the significance of researching hybrid IgG4 k/λ molecules in a rare AD like MuSK-MG.

In the precision medicine era, the validation of novel immunological biomarkers plays a crucial role in the early detection of diagnosis, response to treatment and prognosis. Different reports analyzed serum levels of IgG subclasses in the course of autoimmune diseases, underlying how different subclasses could contribute to pathogenesis of such disorders [29]. In this contest, the unique ability of dynamic IgG4 to exchange half-molecules mediating pathogenesis of MuSK-MG may have diagnostic-prognostic powerful potential. Further analysis is widely desired to completely validate and introduce the assessment of hybrid IgG4 in MuSK-MG clinical management. 

## 4. Materials and Methods

### 4.1. Patients and Samples

For the determination of hybrid IgG4 levels in myasthenic pts, we employed 38 MG sera: 14 from MuSK-MG and 24 from AChR-MG pts. All sera were included in the same batch employed for the analysis of Ig-free light chains [22]. The patient population was referred to the Department of Neurosciences of the Policlinico A. Gemelli in Rome. For each MG pt, serum samples were collected during outpatients’ visits and stored at −20 °C until the analysis was performed. 

Anti-MuSK and anti-AChR Abs were detected by radioimmunoprecipitation assay in our Institution’s central laboratories as routine determinations, using commercial kits for MuSK-Abs (cut-off ≥ 0.05 nmol/L) and for AChR-Abs (cut-off ≥ 0.5 nmol/L) according to the manufacturer’s instructions (RSR Cardiff, UK). 

All patients had generalized MG, as defined by the MGFA clinical classification [30] and were treated according to the accepted guidelines [31]. All patients were under prednisone therapy plus one more immunosuppressant. None of them underwent plasmapheresis, nor received high dose intravenous immunoglobulins, at the time of this study. At the time of treatment, MG severity ranged from IIa to IVb according to the MGFA classification; for 1 out 14 MuSK-MG and for 10 out of 24 AChR-MG pts the MGFA was unknown (see Appendix A).

### 4.2. Analytic Procedure for Hybrid IgG4 k/λ Detection

The analysis was conducted by the Binding Site Group Ltd. of Birmingham (University of Birmingham, UK) in 2016; we sent frozen aliquots of our serum samples to for the determination of hybrid IgG4. All samples were processed anonymously. An enzyme immunoassay (EIA) that enables to measure levels of IgG4 k/λ hybrids directly from serum was developed by The Binding Site Group Ltd. as previously described [17]. We show a schematic illustration of this method in Figure 1B. Briefly, in the first step an anti-total λ light-chain polyclonal antibody (anti-IgG4 λ) was used to capture serum IgG4 λ and IgG4 k/λ molecules, while IgG4 k molecules were not linked (and washed away). Next, only the IgG4 k/λ captured were detected by the addition of an anti-k light-chain horseradish peroxidase (HRP)-conjugated antibodies. Finally, the specific substrate for HRP was added and absorbance was read at 450 nm. This method had an intra assay precision (*n* = 16) of 6.9, 5.6, and 8.7 %CV as demonstrated using samples with a low (25 mg/L), medium (85 mg/L) and high (580 mg/L) IgG4 hybrid concentrations, respectively. The inter assay precision was 14.0, 7.3, 4.3, and 12.3% CV using samples that had median IgG4 k/λ hybrid values of 49.2, 146.6, 148.8, 199.7 mg/L [17]. Total IgG4 molecules were quantified by means of a separate EIA [4,17]. The amounts of IgG4 k/λ molecules in NHS (n:95) appeared to be at ~30% of total IgG4 as reported [17].

### 4.3. Statistical Analysis

Statistical analyses were performed using the software package R (4.0.2 release). Laboratory parameters were tested for normality using a visual inspection of the QQ-Plot, followed by a Shapiro–Wilk test, showing significant deviations from normality. Therefore, non-parametric tests were used in subsequent analyses. Specifically, comparisons between groups were performed with the Wilcoxon Unpaired Two-Sample test. Only *p* values < 0.05 were considered significant. The relationship between selected continuous variables was investigated through univariate linear regression, according to the equation $y=\beta_{0}+\beta_{1}x$. Observed residuals $e_{i}$ from the regression line, defined as the difference between the observed values $y_{i}$ and the predicted one $\hat{y_{i}}$ using the best fit line equation ($e_{i}=y_{i}-\hat{y_{i}}$) are shown to highlight possible unwanted effects in the regression procedure, such as absence of linearity or significant deviation from normality in the residuals associated, for example, with outliers, that may strongly affect the determination of the regression purpose. Normality of residual was investigated using qqplot (data not shown) and a Shapiro–Wilk test.

Pearson correlation coefficients were also evaluated to further investigate the correlation between variables. Strength of correlation was judged using correlation coefficients of >0.70 as strong correlation, 0.30–0.70 as moderate correlation, and <0.3 as weak correlation. 

## Figures and Tables

**Figure 1 ijms-22-09142-f001:**
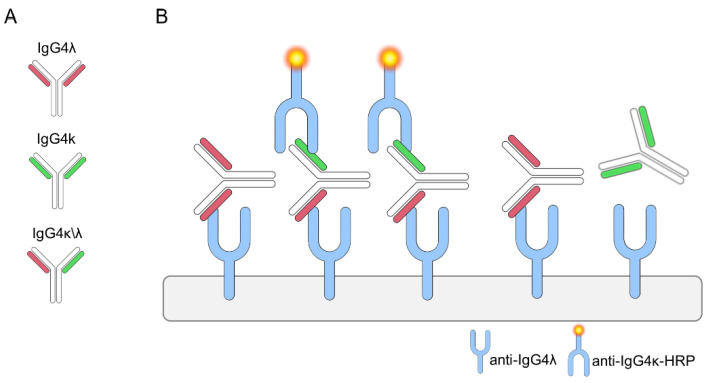
Schematic representation of IgG4 k/λ (**A**) and of the Binding site assay used for the measurement of hybrid levels (**B**).

**Figure 2 ijms-22-09142-f002:**
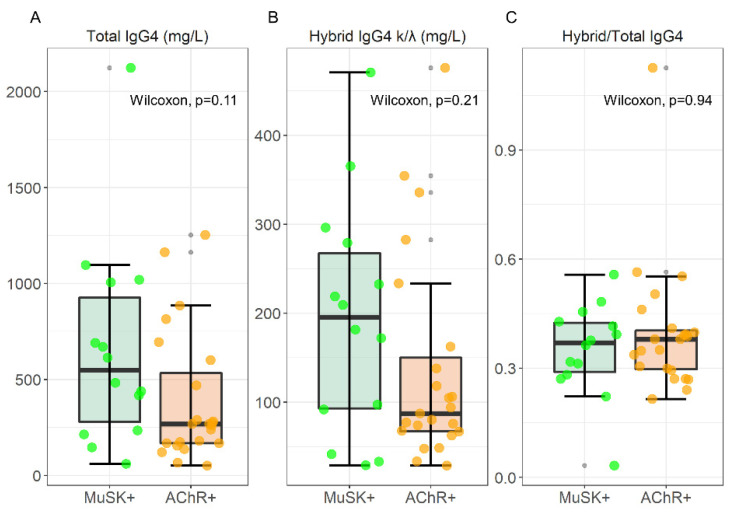
Box-plot analysis of Total IgG4, hybrid IgG4, and Hybrid/Total IgG4 ratio in 14 MuSK and 24 AChR-MG sera. The result of a Wilcoxon unpaired two-sample test is superimposed to each plot.

**Figure 3 ijms-22-09142-f003:**
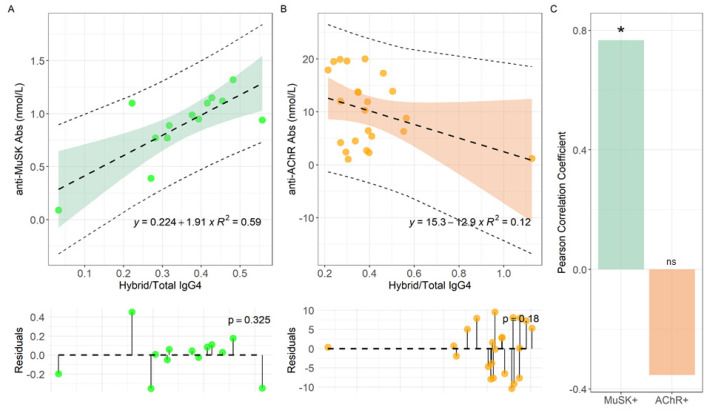
Linear regression analyses of anti-MuSK Abs levels versus Hybrid/Total IgG4 ratio for MuSK-MG (**A**) and AChR-MG (**B**) pts, with the corresponding Pearson correlation coefficient (**C**). Serum samples were analyzed for each MG-pt (n:14 for MuSK-MG and n:24 for AChR-MG). *****: *p* < 0.01. ns: not significant.

**Figure 4 ijms-22-09142-f004:**
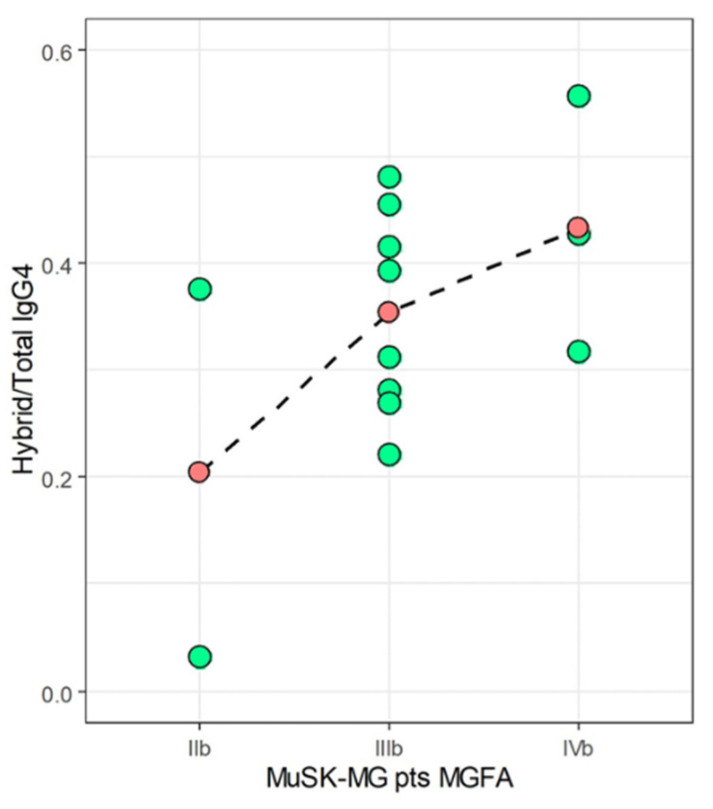
Distribution of Hybrid/Total IgG4 ratios (green-filled dots) at different MGFA scores for MuSK-MG pts (only samples of pts #1-13 were analyzed). Mean values are represented as red-filled dots joined by a dashed line.

**Table 1 ijms-22-09142-t001:** IgG4 determinations in 14 MuSK and 24 AChR-MG sera.

	MuSK-MG			AChR-MG			
IgG4 Determinations	*n*	Median	IQR	*n*	Median	IQR	*p*-Value
Total IgG4 (mg/L)	14	547.92	646.78	24	267.73	366.81	0.11
Hybrid IgG4 (mg/L)	14	195.39	174.39	24	87.16	82.81	0.21
Hybrid IgG4/Hybrid IgG4	14	0.37	0.13	24	0.38	0.11	0.94

## Data Availability

The data presented in this study are available upon reasonable request to the corresponding author.

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
