# Peer review of "Laboratory Investigation of Hybrid IgG4 k/λ in MuSK Positive Myasthenia Gravis"

_ijms, 2021, doi:10.3390/ijms22179142_

Round 1
Reviewer 1 Report
The paper examines the expression and the association of active disease of FAE in the sera of 38 patients with myasthenia gravis of which 14 with antibodies against MUSK and the rest with antibodies against AChR.
Methods
What was the reason for using 2 serum samples from each participant and not one? What is the degree of deviation between the 2 results of the same donor? Did the patients receive the same treatment on both sampling dates?
What were the results if only the first value from each subject was taken?
There seems to be a methodological error in analyzing results that include 2 values from each participant since the parameters must be independent
There is a place to add a control group of a matched healthy individual for age and sex to the tested groups and not just refer to what is defined as the normal by Binding Site Group Ltd, since there may be differences between different populations.
It is recommended to add a figure that explains the methods of analysis of the hybrid k/λ IgG4 and shows an example in a supplementary figure.
Results
The analysis of the correlation between the hybrid/ total IgG4 ratio to the MGFA is unclear. It is not clear whether 2 samples were taken from each patient. To test this correlation? It is more correct to follow patients prospectively and see if the ratio change according to the change in MGFA. Therefore it is suggested to demonstrate the values of each patient between the 2-time points tested in relation to the MGFA grade and to analyzed accordingly
What were the results of assessing the correlation between the hybrid IgG4 and MGFA?
It would be appropriate to test the correlation between anti-MUSK levels and MGFA as these are the pathogenic autoantibody in the disease.
Discussion
Please insert the related reference for the statement: "Since most of pathogenetic MuSK IgG4 consists in hybrid molecules"
It is not clear what is the relevance to the discussion of the treatment with rituximab, which is an effective drug for different types of antibody-mediated diseases
It is strange that the authors chose to end the paper with the issue of the COVID19 pandemic when their study did not deal with that at all. The meaning of the paragraph dealing with this matter is not clear.
It is not clear what the conclusion of the study is?
Minor comment
There is a double numbering of the references and an error starting from ref. 22 and on in the list.
Reviewer 2 Report
We thank the authors for this very interesting manuscript on the role of K/L Igg4 hybrid as predictor of MuSK positive Myasthenia gravis. Our suggestions to improve the acumen of the manuscript as well as increased its breadth are detailed bellow:
1- Line 26-54 : the hybrid antibody are bispecific and bivalent as the have 2 bonding site, not monovalent.
2- Line 27 : it is not clear in the texte that the hybrid have increase pathogenicity.
3- Line 46 : IgG4 are increase after prolong activation of the immune system with or without antigen.
4- Line 58 : IgG4 are considered even considered anti-inflammatory. I think it should be mentioned.
5- Table 1 and 2: What is the second line for every patients ( the white line ). It is not clear if it is a duplicate or something else. A legend should be built.
6- Line 159: The nature of the fitting residues should be described as most physician reading this manuscript will not have such advance statistical knowledge.
7- Line 180: It is not significant, I do not think it can be claimed that is strongly suggest any association. This should be tune down.
8- Line 207: This increase is not supporter by the data; not significant.
9- Line 248: The link with COVID-19 is not link with the text and the results presented. I think this part add confusion to the text. It should either better linked to the text or simply removed.
Reviewer 3 Report
The paper describes hybrid IgG4involvement in progressive disease in myasthenia patients with anti-MuSK ab disease.
There are several grammatical errors that need adjusting such as line 55 and miss to form change to fail to form
Line 71 for the most majority change to part
Line 76 IgG4 do undergo change to undergo
Line 77 would rather delete rather
Line 87 IgG4 level changes to levels
Line 94 routinely change to routine
line 213 Abs are searched change to studied
Line 218 contest change to context
Line 225 On the contrary change to in contrast
Round 2
Reviewer 1 Report
The most important observation deals with the correlation of hybrid/total IgG4 ratio and MGFA . The author did not changed their analysis and left the 2 measurement from each of the patients tey had MGFA score , this matter is a problem in the correct analysis of the results. The author could not add a group of healthy controls . This things interfere with the quality of the paper and prevent me from recommending it acceptance.
Author Response
Query #1
The most important observation deals with the correlation of hybrid/total IgG4 ratio and MGFA. The author did not changed their analysis and left the 2 measurement from each of the patients tey had MGFA score, this matter is a problem in the correct analysis of the results.
Response #1:
Accordingly to the Reviewer we changed Figure 4: in the new one, we show the distribution of Hybrid/Total IgG4 ratios at different MGFA scores for 13 out 14 MuSK-MG pts who were evaluated for MGFA at the time of withdrawal, analyzing only the first sample of each patient (samples #a). The very limited number of samples prevents us from any statistical consideration; we can only observe a progressive increase of the titer with the increasing of MGFA score. Specifically, IIb MuSK-MG pts had the lowest mean ratio of 0.20, IIIb pts an inter-mediate mean value of 0.35, and IVb pts showed the highest mean value of 0.43. These results are suggestive that the hybrid/IgG4 ratio correlates with disease severity, but this finding deserves a better in-depth study in a larger sample size to be confirmed. To better underline that this observation is not supported from any statistical significance we also changed “strengthen” into “validate” (line 241 of Discussion).
Query #2
The author could not add a group of healthy controls.
Response #2:
We retain that the issue regarding the recruitment of a control group has been adequately addressed through the citation of data in the literature (ref #4: Young E.; Lock E.; Ward D.G.; Cook A.; Harding S.; Wallis G.L. Estimation of polyclonal IgG4 hybrids in normal human serum. Immunology 2014, 142, 406-13; ref #17: Cook A; Taylor D; Sims D; Harding S; Wallis G. Direct measurement of polyclonal IgG4 fab-arm exchange in serum, Clinical Chemistry 2016, 62, No. 10, Supplement, S93).

Round 3
Reviewer 1 Report
Indeed the results are now more compatible and so it can be seen that they are weaker and not significant when are analyzed adequately. This indicates a significant weakness of the study and it is not possible to use 2 samples from the same participants in such analyses to overcome this weakness. Figures 2 and 3 still show a pair of results from the same participants. Moreover, being familiar such measurments, it is important that at the same time the researcher will check a group of healthy donors and not just rely on previous reports.
Author Response
REVIEWER 1
Indeed the results are now more compatible and so it can be seen that they are weaker and not significant when are analyzed adequately. This indicates a significant weakness of the study and it is not possible to use 2 samples from the same participants in such analyses to overcome this weakness.
Figures 2 and 3 still show a pair of results from the same participants.
Moreover, being familiar such measurments, it is important that at the same time the researcher will check a group of healthy donors and not just rely on previous reports.
REPLY to Reviewer 1
We gratefully thank the Reviewer that, through the critical comments raised, allowed us to ameliorate the quality of our manus.
Comment #1: “Figures 2 and 3 still show a pair of results from the same participants”.
RESPONSE:
We had already corrected the Figure 2 in the precedent revised version. However, according to the Reviewer, in this new revision we also changed Figure 3, showing only the first result of each participant. We totally accepted the Reviewer objection, and we deleted any references two the 2 samples also from the main text.
For this reason, we also decide to modify Table 1 and 2 showing only one sample for each patient and to remove these two Tables from the manuscript, showing them as Supplementary Tables S1 and S2. Accordingly, the old Table 3 becomes the new Table 1. We totally agree with the Reviewer about the major limitation of our study due to the small sample size that we stated in the discussion from the first submission; however, we are confident about the validity of our main effort to translate into the clinical practice the significance of researching hybrid IgG4 k/λ molecules in a rare AD as MuSK-MG.
Comment#2: “Moreover, being familiar such measurments, it is important that at the same time the researcher will check a group of healthy donors and not just rely on previous reports.”
RESPONSE:
We thank the Reviewer for this comment that induced us to modify the discussion (from line 219 to line 242 and that we here report, remarking the major points:
The FAE phenomenon naturally occurs in vivo taking in account of bispecific IgG4s also in normal human serum. Data from literature showed that hybrid/total IgG4 ratio in NHS is assessed at approximately 30%[4,17]. Analysis on serum samples from a range of IgG4-related AD were also conducted giving different results: a lower hybrid/total IgG4 ratio has been described in samples from Pemphigus/Pemphigoid and Autoimmune Pancreatitis while a ratio similar to NHS in Primary Sclerosing Cholangitis [17].
Here we show our findings of a pilot laboratory investigation of hybrid IgG4 k/λ assessment in MuSK-MG sera. We observed that, although an increased (not significant) in IgG4 total levels, the hybrid/total IgG4 ratio of 0.37 [0.13] is comparable to AChR-MG, the not IgG4-mediated form (0.38 [0.11]), and reflects the occurrence reported in literature for NHS. Since only MuSK-MG is the IgG4-mediated MG form, we suggest that is not the total amount of hybrids per se the matter of the fact (and if it is accordant or discordant from NHS levels) but if hybrids (that are exclusively IgG4 molecules) are engaged in pathogenetic immune-reactivity or not. For this evaluation we retain more useful the comparison between the two MG subtypes, the hybrid IgG4 mediated (MuSK-MG) and the not hybrid IgG4-mediated (AChR-MG). When we compared the hybrid/total IgG4 ratios with anti-MuSK Abs titers we found a strong significant correlation (ρ=0.77, p=0.002) indirectly confirming that the most part of hybrid IgG4 molecules are engaged in the anti-MuSK immune reactivity, mediating pathogenicity (Figure 3A). Instead, when we analyzed the hybrid/total IgG4 ratios in relation to anti-AChR titers we found an inverse negative correlation, that is coherent with the assumption that AChR-MG is not an IgG4 mediated subtype of MG (Figure 3B). The observation that the mean value of hybrid/total IgG4 ratio in MuSK-MG increases along with disease severity according to the MGFA score further reinforces our previous result (Figure 4).
We hope that now our paper will be considered suitable for publication on IJMS.
